# Current Medical Care Situation of Patients in Germany Undergoing Pressurized Intraperitoneal Aerosol Chemotherapy (PIPAC)

**DOI:** 10.3390/cancers14061443

**Published:** 2022-03-11

**Authors:** Philipp Horvath, Can Yurttas, Isabella Baur, Christoph Steidle, Marc André Reymond, Paolo Nicola Camillo Girotti, Alfred Königsrainer, Ingmar Königsrainer

**Affiliations:** 1Department of General, Visceral and Transplant Surgery, University Hospital Tübingen, Hoppe-Seyler-Str. 3, D-72076 Tübingen, Germany; philipp.horvath@med.uni-tuebingen.de (P.H.); can.yurttas@med.uni-tuebingen.de (C.Y.); isabella.baur@med.uni-tuebingen.de (I.B.); christoph.steidle@med.uni-tuebingen.de (C.S.); marc.reymond@med.uni-tuebingen.de (M.A.R.); alfred.koenigsrainer@med.uni-tuebingen.de (A.K.); 2Department of General, Visceral and Thoracic Surgery, Academic Teaching Hospital Feldkirch, Carinagasse 47, 6807 Feldkirch, Austria; paolo.girotti@lkhf.at

**Keywords:** gastric cancer, PIPAC, peritoneal metastases, cost effectiveness, histological, regression

## Abstract

**Simple Summary:**

The treatment of peritoneal surface malignancies has evolved differently from other fields in surgical oncology in the past decades and modern approaches were introduced into daily-routine. Curative as well as palliative surgical treatment options redefined the fate of patients with this aggressive disease. Nowadays, patients with a limited peritoneal tumor spread of colonic origin have comparable survival rates compared with colorectal liver and pulmonary metastases. After the failure of systemic chemotherapy, the application of PIPAC revolutionized the palliative treatment for these patients and recent results show a high histological response rates, an improvement in quality-of-life and, furthermore, a cost-reduction compared to systemic chemotherapy. The application of PIPAC in other disease-settings (neoadjuvant/adjuvant) needs to be defined in further clinical trials.

**Abstract:**

Objective: Tailored approaches in gastrointestinal oncology have been more frequently introduced in past years and for patients with peritoneal metastases. This article attempts to overview the current strategies in surgical gastrointestinal oncology, with a focus on gastrointestinal peritoneal metastases. Methods: In 2019, all patients undergoing PIPAC therapy in Germany were retrospectively analyzed regarding morbidity and in-hospital mortality rates. Furthermore, patients with chemotherapy-refractory peritoneal metastases from gastric cancer undergoing PIPAC-therapy at our institution were analyzed. Results: In 2019, 534 patients received PIPAC treatment in german hospitals. The in-hospital mortality rate was 0%. In total, 36 patients suffered from postoperative complications (8%). From April 2016 to September 2021, a total of 44 patients underwent 93 PIPAC applications at our institution. The non-access-rate was 0%. The median PRGS was two (range, 1–4). Eleven patients (44%) showed histologically stable disease, whereas six patients (24%) showed histological regression. Median survival, calculated from the date of the first PIPAC application, was 181 days (range, 43–636 days). Conclusions: PIPAC is a safe and feasible procedure with a low in-hospital morbidity and mortality. Furthermore, PIPAC in the palliative and chemorefractory setting and is an appealing approach for patient management in the future.

## 1. Introduction

Tailored modern approaches have very much altered therapy algorithms for a variety of gastrointestinal malignancies. Additional mutational analysis of tumor specimens for the detection of driver mutations, i.e., KRAS and BRAF-mutations, expanded the options for systemic chemotherapy in metastatic colorectal cancer patients and optimized treatment outcomes. Modern approaches for the treatment of peritoneal metastases evolved considerably during the past two decades. From an inoperable and palliative situation, with the only therapeutic option being systemic chemotherapy, nowadays, a curative treatment can be provided for a selected patient subset.

The applied therapy approaches not only changed for the curative but also for the palliative setting. In the curative setting, a variety of patient- and tumor-related parameters have to be fulfilled, before offering cytoreductive surgery and hyperthermic intraperitoneal chemotherapy (HIPEC) to the patients. A favorable PCI (peritoneal cancer index) (colorectal: <16 [1], gastric: <6–10 [2], ovarian: not defined yet [3,4,5]), a favorable histology (signet-ring histology is considered a relative contraindication for CRS and HIPEC [6,7]), and response to systemic chemotherapy should be present. These parameters should allow for a proper patient selection and in combination with a structured perioperative complication management, acceptable morbidity- and mortality-rates should be achievable.

On the other hand, the treatment options in the palliative setting were very much revolutionized by the implementation of PIPAC (Pressurized intraperitoneal aerosol chemotherapy). Recent data from controlled trials suggest that PIPAC seems to improve objective tumor response, survival and quality of life in patients with peritoneal metastasis of gastrointestinal and gynecological primary tumors after exploitation of systemic chemotherapeutic options [8]. A recent meta-analysis from Alyami and colleagues, summing up more than 1300 PIPAC applications in more than 600 patients, clearly outlined the safety (adverse events (CTCAE 4.0) Grade 3: 10.4%; Grade 4: 1.7%) and efficacy (histological regression in the per-protocol cohort between 57% and 91%) of this drug-delivery system [8]. Despite these overwhelming results in the palliative treatment setting, PIPAC has not yet found its way into german guidelines recommendations for the treatment of peritoneal metastases of gastrointestinal and gynecological primary tumors. The only exception is the treatment of gastric peritoneal metastasis within a clinical trial. Nevertheless, the german DRG- (Diagnosis Related Groups) system introduced a unique OPS (German Procedure) code (5-549.b) for PIPAC application.

In this manuscript, we summarise the current medical care situation of patients under PIPAC-therapy and highlight the efficacy of PIPAC for the treatment of peritoneal spread of gastric origin.

## 2. Materials and Methods

Data on all patients treated since the introduction of the unique OPS-code for PIPAC were obtained from the nationwide German diagnosis-related group (DRG) statistics hosted by the German Federal Statistics Office. Data management strictly followed German data protection regulations. Patients with OPS (German procedure codes) code 5-549.b (Pressurized intraperitoneal aerosol chemotherapy), which was introduced 2019, were included, and analysis was restricted to patients with complete data records. The OPS registry is a modified version of the International Classification of Procedures in Medicine (ICPM). The procedure had to be performed in a German hospital.

Furthermore, all patients with peritoneal metastasis of gastric origin who underwent PIPAC application at our institution from 2016 to 2021 were evaluated regarding defined patient- and treatment related factors. The data ware extracted from a prospective database.

Patients scheduled for PIPAC treatment had to fulfill the following conditions: (1) histologically proven PM of gastric cancer; (2) progression of PM under or after systemic chemotherapy; (3) positive vote after evaluation by our multidisciplinary tumor board; (4) Karnofsky Index ≥ 60%; (5) no clinical or radiological signs of bowel obstruction; and (6) no extraperitoneal disease.

### 2.1. Ethical and Regulatory Framework

The international PIPAC patient registry was approved by the Ethics Committee, Ruhr-University Bochum, on 11 January 2016 (reference 15-5280), and by the data protection officer of the State of North Rhine-Westphalia, Germany. Each patient gave his/her written informed consent both for the PIPAC procedure and for data-storage management and analysis. The registry is hosted by an independent quality control organization (AnInstitut für Qualitätssicherung in der Operativen Medizin gGmbH, Otto-von-Guericke Universität Magdeburg) [9]. Since 2020, the international PIPAC registry has been hosted by the University Hospital Odense, Denmark. Although cisplatin, doxorubicin, and/or oxaliplatin is routinely used in clinical practice worldwide for locoregional therapy in peritoneal disease and has been the object of multiple randomized controlled trials [10,11,12,13,14,15,16,17], none of these drugs are currently approved for intraperitoneal delivery. Therefore, PIPAC was applied “off-label”.

### 2.2. Technical Aspects of PIPAC Application

The PIPAC application was performed as described in other publications [18,19,20]. Access to the abdominal cavity was usually obtained via Veress needle at Palmer’s points. If application and maintenance of the pneumoperitoneum was not possible, a mini-laparotomy was performed. Non-access was defined as the impossibility to safely access the abdominal cavity with two trocars in order to safely perform peritoneal biopsies and safe aerosolization of the chemotherapeutic agents. To start, the first 5 mm trocar was placed. Next, the 12 mm trocar was introduced opposite to the 5 mm trocar tu guarantee an optimal visualization of the micropump. Then, the PCI-index according to Sugarbaker was evaluated [21].

Peritoneal biopsies were taken from all four abdominal quadrants whenever possible. No biopsies were taken from the peritoneal at the diaphragms. Present ascites were completely removed and the amount was documented. No adhesiolysis was performed. The PIPAC Micropump (MIP1 Micropump/Capnopen (Reger Medizintechnik, Rottweil, Germany)) was inserted into the 12 mm trocar [22,23]. Prior to aerosolization, a standardized checklist was processed. Zero flow of CO_2_ was verified. The chemotherapeutic agents, doxorubicin at a dose of 1.5 mg/m^2^ body surface area (BSA), followed by cisplatin at a dose of 7.5 mg/m^2^ BSA were aerosolized. The toxic aerosol was maintained at 12 mmHg for 30 min at 37 °C. Finally, the aerosol was evacuated from the abdominal cavity using a Closed Aerosol Waste System (CAWS). Trocars were then retracted and the laparoscopy terminated. The trocar insertion site were closed. Previously reported safety standards for PIPAC therapy were followed [24].

### 2.3. Histological Regression Analysis

Peritoneal specimens were analyzed and staged according to the Peritoneal Regression Grading Score (PRGS) [20]. The scale ranges from 1 to 4 and is based on histological features of regression, including fibrotic changes, necrosis, and the presence of acellular mucin deposits. PRGS 1 (complete regression) is defined as no tumor cells visible; PRGS 2 (major response) as regressive changes predominant over tumor cells; PRGS 3 (minor response) as a predominance of tumor cells but minimal regressive changes detectable, and PRGS 4 (no response) as no regressive changes present.

The mean and worst PRGS-values were documented. PRGS evaluation at the first PIPAC cycle reflects systemic chemotherapy-induced regression. 

## 3. Results

### 3.1. German PIPAC-Data

From January to December 2019, a total of 534 patients received PIPAC treatment in german hospitals. There were 219 males and 315 females. Patient age ranged from 20 to 25 years to patients 85 and 90 years (Table 1). The largest proportion of patients (*n* = 93; 17%) were between 55 and 60 years of age.

Patients who received PIPAC treatment had the following primary diagnosis according to ICD-10 classification (C16 (gastric cancer); C18 (colon cancer); C45 (mesothelioma); C48 (other tumors of the peritoneum); C56 (ovarian cancer) and C78). The vast majority of patients (*n* = 452; 85%) had C78 (secondary tumors originating from the small bowel, rectum, liver, biliary tree system, pleura and lung) as their primary diagnosis. The in-hospital mortality rate was 0%. In total, 36 patients suffered from postoperative complications (*n* = 36; 8%) (Table 1). The most common postoperative complications were pleural effusion (*n* = 10) and postoperative paralytic ileus (*n* = 13). All patients had C78 as primary diagnosis. None of the patients had to be re-operated.

### 3.2. Single Center Data

From April 2016 to September 2021, a total of 44 patients (m:f = 22:22) with a median age of 50 years (range, 24–79 years) underwent 93 PIPAC applications at our institution. The vast majority of patients (*n* = 19) underwent only one PIPAC application, whereas 13 patients underwent two PIPAC cycles, five underwent three PIPAC cycles, two underwent four PIPAC cycles and five underwent five PIPAC cycles. The median Karnofsky index was 80% (range, 60–100%). All but one patient received cisplatin and doxorubicin as chemotherapeutic agents. Due to an anaphylactic reaction to platin, one patient received doxorubicin monotherapy. In 27 patients (61%) synchronous peritoneal metastases occurred and in 17 patients (39%) metachronous peritoneal metastases occurred. Non-access-rate was 0% and in total four patients (9%) had additional surgical procedures (adhesiolysis). Median operative time was 101 min (range, 43–210 min). Median PCI was 27 (range, 1–39). The median amount of ascites was 80 mL (range, 0–8000 mL). Median PRGS was 2 (range, 1–4). For 25 patients who received more than one PIPAC application, the histological response rate were calculated. Eleven patients (44%) showed histologically stable disease whereas six patients (24%) showed histological regression. Eight patients (32%) displayed progressive disease. Median survival, calculated from the date of the first PIPAC application, was 181 days (range, 43–636 days). Data are summarized in Table 2.

## 4. Discussion

The evolution of surgical and chemotherapeutic treatment options for patients with peritoneal surface malignancies was surely one of the most remarkable developments in the past decade. In no other field of surgical oncology was a shift from an inoperable and palliative setting to a situation where a certain subset of patients can be treated with a curative intent performed. The application of heated intraperitoneal chemotherapy after the resection of all visible tumor deposits on the peritoneal surface leads to a significant improvement in survival and for a variety of primary and secondary peritoneal tumors (i.e., peritoneal mesothelioma and pseudomyxoma peritonei), CRS and HIPEC represent the treatment of choice. Furthermore, recent data indicate quite clearly a palpable benefit of additional HIPEC for patients with primary ovarian cancer [13]. A recent randomized trial assigned patients to undergo secondary cytoreductive surgery and then receive platinum-based chemotherapy or to receive platinum-based chemotherapy alone in the recurrent disease setting with a platin-free interval of more than 6 months. All patients had to fulfill the DESKTOP-I criteria. Surgery and additional systemic chemotherapy resulted in a statistically improved overall survival (HR 0.75; *p* = 0.02) [25]. Moreover, the treatment options for patients with inoperable peritoneal surface malignancies, not eligible for CRS and HIPEC, expanded, and once again in no other field of surgical oncology is a surgical treatment option available for this subset of patients. Even though some major cornerstones of PIPAC have not yet been defined, so far, the data quite clearly indicate a clinical benefit for patients after failure of systemic chemotherapy [8]. The first PIAPC application in the world was performed in Germany in 2011. Since then, a number of publications have been completed and right now more than 20 clinical trials are recruiting patients in different treatment settings and for different tumor etiologies. Similar to every young treatment introduced to the broad atrium of medicine, PIPAC invited a lot of skepticism and criticism. The recent work from Alyami et al. [8] succinctly demonstrated that in more than 1810 PIPAC applications in 838 patients, grade 3 and 4 complications were very rare (grade 3: 10.7%; grade 4: 1.7%). This is in line with almost every publication discussing PIPAC. It is a safe and very well-tolerated treatment option for patients in this disease setting. The data presented in this manuscript are in agreement with this. In 2019, the German DRG-system introduced a new OPS-code for PIPAC application and since then all data have been recorded. The national data from 2019 from all german hospitals performing PIPAC showed 0% in-hospital mortality and a morbidity-rate of only 8%. The data set registered at the “Statistisches Bundesamt” did not include the severity of each complication, so we cannot comment on the true rate of grade 3 or 4 complications. The most common postoperative complications were pleural effusion (*n* = 10) and postoperative paralytic ileus (*n* = 13). This is quite comprehensible, because after the evacuation of ascites in patients with hypalbuminemia, the occurrence of pleural effusion is common. PIPAC induces a sort of chemical peritonitis, specifically affecting the small bowel. Thus, the appearance of postoperative ileus is often encountered in the postoperative setting. In general, these clinical conditions can be treated very easily.

Unfortunately, the analyzed data set did not provide information regarding patient survival under PIPAC treatment. Recent data show that PIPAC is effective in a variety of primary (mesothelioma) and secondary (gastric, colorectal, pancreas, biliary tract) peritoneal surface malignancies. Furthermore, the objective clinical response rates exceed those achieved by systemic chemotherapy by far. The best available review on PIPAC [8] showed very promising results for patients with refractory isolated peritoneal metastasis indeed. An objective clinical response of 62–88% was reported for patients with ovarian cancer (median survival of 11–14 months), 50–91% for gastric cancer (median survival of 8–15 months), 71–86% for colorectal cancer (median survival of 16 months), and 67–75% (median survival of 27 months) for peritoneal mesothelioma. These overwhelming data must be compared with patients under best-supportive-care and document very impressively the role of PIPAC in this patient subset. Our own data showed that in 68% of patients who received more than one PIPAC application, a stable and improved disease setting was observed. These results substantiate the rationale for PIPAC application in patients with metastatic gastric cancer with a “peritoneal-only” site of disease. Furthermore, the available evidence for the efficacy of currently recommended systemic therapies in the second-line situation is still unsatisfactory and the majority of patients who receive second-line treatment fail to achieve a response [26]. Apart from the clinical benefit for patients under PIPAC therapy, a recent analysis showed that the application of PIPAC in patients with peritoneal metastases of gastric cancer is a cost-effective strategy in the first as well as in the second-line setting [26]. These results are of utmost importance, demonstrating the superiority of PIPAC over systemic chemotherapy in these patients as far as clinical and cost-efficacy is concerned.

Nationwide as well as our single-center data showed that PIPAC is a very safe and feasible procedure and the implementation of PIPAC in the context of repetitive systemic chemotherapy application constitutes no major problem. In our series, we encountered 0% grade IV complications and the non-access rate was 0%, which is in line with other groups [27]. Tempfer et al. reported a laparoscopic non-access rate of 11/64 (17%) [28].

The drug combination (cisplatin/doxorubicin) used has a very low potential for triggering postoperative complications. On the other hand, some severe complications (acute respiratory distress syndrome (ARDS) and severe peritoneal fibrosis) were described after PIPAC application with oxaplatin [29,30].

The role of PIPAC in other disease settings (curative-adjuvant/neoadjuvant or palliative-first line) are not yet properly defined. Expected data from the Odense-group analyzing the efficacy of adjuvant PIPAC application in patients with high-risk colon cancer are eagerly awaited. [31]. The authors expect an absolute risk reduction of 15% regarding the development of peritoneal metastases in high-risk colon cancer patients.

Lastly, recent research focused mainly on technical modifications of the described PIPAC technique. These modifications concerned the application time in the context of electrostatic_PIPAC (e-PIPAC), drug concentrations in a dose-escalation model and the aerosolization of oncolytic adenoviruses [32,33,34]. Data indicate that e-PIPAC is non-inferior to “classical” PIPAC application, with a steady-state of 30 min after aerosolization of the chemotherapeutic substances [32]. Further results from larger cohorts are needed before the true benefit of e-PIPAC can be defined. Robella et al. conducted a dose-escalation study for oxaliplatin, cisplatin and doxorubicin and defined “new” dosages, which need to be assessed in further phase I-II trials [33]. The intraperitoneal administration of adenoviruses via the drug-delivery-device “PIPAC” has been proven to be feasible in the rat model but data are not yet sufficient to undertake in vivo application [34].

## 5. Conclusions

In conclusion, PIPAC is a safe and effective treatment option for patients with peritoneal metastases of gastrointestinal and gynecological origin. The excellent safety profile is accompanied by a superiority regarding costs compared to systemic chemotherapy. The surgical morbidity is negligibly small and the sequence of systemic chemotherapy and PIPAC is not hindered. Future research should be focused on two cornerstones. The first is further in vitro and in vivo phase-I trials to evaluate efficacy of other drugs and drug formulations, and the second is randomized clinical phase-III trials comparing PIPAC and standard-of-care systemic chemotherapy in the first-line for peritoneal metastases.

These future results will define the true impact of PIPAC in the context of peritoneal metastases.

## Figures and Tables

**Table 1 cancers-14-01443-t001:** Patient data of the DRG-cohort.

Parameter	Cohort (*n* = 534)
**Sex (male) % (*n*)**	41 (219)
**Tumor etiology % (*n*)**	
Colon Ovarian Gastric Mesothelioma Not defined (C78)	1 (4) 1 (3) 4 (20) 6 (34) 88 (473)
**Complication rate % (*n*)**	
Total	8 (36)
**Complication type % (*n*)**	
Bleedina anemia Pleural effusion Respiratory insufficiency Paralytic ileus Renal failure	1 (5) 2 (10) 1 (5) 3 (13) 1 (3)
**In-hospital mortality**	0%

**Table 2 cancers-14-01443-t002:** Patients- and treatment-related parameters of the single center data (PCI = peritoneal cancer.

Parameter	Cohort (*n* = 44)
Median Age (range)	50 (27–77)
Sex (male) % (*n*)	50 (22)
Tumor etiology % (*n*)	
Gastric	100 (44)
Median PCI (range)	27 (1–39)
Disease setting % (*n*)	
Synchronous Metachronous	61 (27) 39 (17)
HIPEC compound % (*n*)	
Cisplatin/Doxorubicin Doxorubicin	97 (43) 3 (1)
Karnofsky-Index (%) median (range)	80 (60–100)
Non-access rate % (*n*)	0 (0)
>2 PIPAC applications % (*n*)	57 (25)
Adhesiolysis % (*n*)	9 (4)
Ascites (ml) median (range)	80 (0–8000)
PRGS median (range)	2 (1–4)
Histological regression % (*n*)	
No regression Stable Regression	32 (8) 44 (11) 24 (6)
Operative time (min) median(range)	101 (43–210)
Survival (days)	181 (43–636)
In-hospital mortality	0%

## Data Availability

Data (DRG-data) available in a publicly accessible repository that does not issue DOIs. Publicly available datasets were analyzed in this study. This data can be found here: https://www.destatis.de/DE/Home/_inhalt.html (accessed on 8 January 2022). Data supporting results (single center data) are harbored by an in-hospital database. Regulatory issues do not allow to provide a link to analyzed data sets.

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
