# Peer review of "Current Medical Care Situation of Patients in Germany Undergoing Pressurized Intraperitoneal Aerosol Chemotherapy (PIPAC)"

_cancers, 2022, doi:10.3390/cancers14061443_

Round 1
Reviewer 1 Report
The paper is methodologically correct and of interest. The series presented wide and compete. Only one question: why was cisplatin used and not oxaliplatin?
I recommend the publication of the paper
Author Response
Thank you very much for this interesting and important question. We use OX-PIPAC in patients with colorectal derived peritoneal metastases. In gastric cancer we use for PIPAC the same drug combination as used in HIPEC. The combination of Cisplatin and Doxorubicin has been proven to be very effective in gastric cancer cell lines in vitro. Furthermore, Cisplatin showed a crucial benefit in heated intraperitoneal chemotherapy in ovarian cancer.

Reviewer 2 Report
It is necessary to congratulate the Authors for this first analysis of the german data concerning PIPAC.
I do not have major concerns but I strogly recommend a robust revision of the English language and a great effort in order to facilitate the reader's task (for example: "acces rate" should be defined in material and methods)
Author Response
Thank you for your comments.
We are going to perform a revision of the manuscript concerning the English language.
The following paragraph was added in the part “material and methods”.
“Non-access was defined as the impossibility to safely access the abdominal cavity with two trocars in order to safely perform peritoneal biopsies and safe aerolization of the chemotherapeutic agents.”

Reviewer 3 Report
The authors present German nation-wide data and their own local data on intraperitoneal chemotherapy of peritonela carcinomatosis from various origins applied via PIPAC regarding feasibility and safety. This is a excellent summary of real-world data confirming the safety and feasibility of this palliative treatment.
I have the following remarks:
1) Please add a summary from literature reports regarding adverse events such as intestinal fibrosis and severe pain evemts after PIPAC (e.g. Thibaudeau E, Brianchon C, Raoul JL, Dumont F. Acute respiratory distress syndrome (ARDS) after pressurized intraperitoneal aerosol chemotherapy with oxaliplatin: a case report. Pleura Peritoneum. 2021 Oct 5;6(4):167-170 and Graversen M, Detlefsen S, Pfeiffer P, Lundell L, Mortensen MB. Severe peritoneal sclerosis after repeated pressurized intraperitoneal aerosol chemotherapy with oxaliplatin (PIPAC OX): report of two cases and literature survey. Clin Exp Metastasis. 2018 Mar;35(3):103-108.)
2) Please comment on how many PIPAC patients in the registry and in your own series were able to undergo secondary cytoreductive surgery after PIPAC after initial failure to undergo surgery befor PIPAC.
3) Please comment on the problem that no phase III trial data are available despite 10 years of PIPAC research.
4) Please comment on the diferential performance of PIPAC with different chemo substances in different tumor entities.
5) Please add one paragraph on the literature data assessing modifications of application length, pressure settings, chemo concentration, nozzle positioning, etc. on the treatment effect regarding penetration depth, peritoneal coverage, and anti.tumor effect.
6) Please discuss the most important research needs which should be addressed in the future in this dynamic field.
Author Response
Reviewer#3:
The authors present German nation-wide data and their own local data on intraperitoneal chemotherapy of peritonela carcinomatosis from various origins applied via PIPAC regarding feasibility and safety. This is a excellent summary of real-world data confirming the safety and feasibility of this palliative treatment.
I have the following remarks:
1) Please add a summary from literature reports regarding adverse events such as intestinal fibrosis and severe pain evemts after PIPAC (e.g. Thibaudeau E, Brianchon C, Raoul JL, Dumont F). Acute respiratory distress syndrome (ARDS) after pressurized intraperitoneal aerosol chemotherapy with oxaliplatin: a case report. Pleura Peritoneum. 2021 Oct 5;6(4):167-170 and Graversen M, Detlefsen S, Pfeiffer P, Lundell L, Mortensen MB. Severe peritoneal sclerosis after repeated pressurized intraperitoneal aerosol chemotherapy with oxaliplatin (PIPAC OX): report of two cases and literature survey. Clin Exp Metastasis. 2018 Mar;35(3):103-108.)
Thank for this very important information. Indeed OX-PIPAC is capable of inducing intestinal fibrosis, pancreatitis and severe abdominal pain. In the adjuvant-PIPAC-OX trial for high-risk colon cancers by Graversen et al. there were some dose-limiting effects. These complications are so far not described in patients with Cis/Dox PIPAC but as you stated this literature reports should be mentioned in the discussion.
Added paragraph in the “discussion” part:
“The used drug combination (cisplatin/doxorubicin) has a very low potential for triggering postoperative complications. On the other hand, some severe complications (acute respiratory distress syndrome (ARDS) and severe peritoneal fibrosis) were described after PIPAC application with oxaplatin [29, 30].”
Added references:
- Graversen M, Detlefsen S, Pfeiffer P, Lundell L, Mortensen MB. Acute respiratory distress syndrome (ARDS) after pressurized intraperitoneal aerosol chemotherapy with oxaliplatin: a case report. Pleura Peritoneum. 2021 Oct 5;6(4):167-170
- Graversen M, Detlefsen S, Pfeiffer P, Lundell L, Mortensen MB. Severe peritoneal sclerosis after repeated pressurized intraperitoneal aerosol chemotherapy with oxaliplatin (PIPAC OX): report of two cases and literature survey. Clin Exp Metastasis. 2018;35:103-108.
2) Please comment on how many PIPAC patients in the registry and in your own series were able to undergo secondary cytoreductive surgery after PIPAC after initial failure to undergo surgery before PIPAC.
Thank for this important remark.
As far as the registry concerned I do not know the correct number of patients who were able to undergo secondary CRS and HIPEC. At our institutions so far only one patient was eligible for this treatment algorithm, a patient with peritoneal epitheloid mesothelioma.
3) Please comment on the problem that no phase III trial data are available despite 10 years of PIPAC research.
At clintrial.gov right now 34 trial are registered, mostly phase I-II trials. For a young treatment option 10 years is a very short time period. In the early stages of PIPAC there was a lot of lack of acceptance from the oncologist. In the last 2-3 years some very convincing results were provided, the PIPAC registry was relaunched in Odense, Denmark and I am convinced that in the next couple of years results of Phase III trials will be available.
4) Please comment on the diferential performance of PIPAC with different chemo substances in different tumor entities.
The use of oxalplatin in colorectal cancer derived peritoneal metastases is analogous to that in systemic chemotherapy. Randomly, the combination of Cis/Dox was chosen years ago for HIPEC treatment and showed clinical efficacy. The penetration depth of Dox during PIPAC is very convincing.
But unfortunately, we have to state that the choice for one or the other drug in intraperitoneal chemotherapy lacks some basic research results. That’s for sure.
5) Please add one paragraph on the literature data assessing modifications of application length, pressure settings, chemo concentration, nozzle positioning, etc. on the treatment effect regarding penetration depth, peritoneal coverage, and anti.tumor effect.
Thank you for that comment. We added a paragraph in the Discussion.
“At last recent research focused mainly on technical modifications of the described PIPAC technique. These modifications concerned application time in the context of electrostatic_PIPAC (e-PIPAC), drug concentrations in a dose-escalation model and aerolization of oncolytic adenoviruses [32, 33, 34]. Data indicate that e-PIPAC is non-inferior to “classical” PIPAC application with a steady-state of 30 minutes after aerolization of the chemotherapeutic substances [32]. Further results from larger cohort are needed before the true benefit of e-PIPAC is evident. Robella et al. conducted a dose-escalation for oxaliplatin, cisplatin and doxorubicin and defined “new” dosages, which need to be assessed in further phase I-II trials [33]. The intraperitoneal administration of adenoviruses via the drug-delivery-device “PIPAC” has been proven to be feasible in the rat model but so far data are not sufficient yet to undertake in-vivo application [34]. “
Furthermore we added 3 new references.
“Tate SJ, Van de Sande L, Ceelen WP, Torkington J, Parker AL. The Feasibility of Pressurised Intraperitoneal Aerosolised Virotherapy (PIPAV) to Administer Oncolytic Adenoviruses. Pharmaceutics. 2021 Nov 30;13(12):2043.
Robella M, De Simone M, Berchialla P, Argenziano M, Borsano A, Ansari S, Abollino O, Ficiarà E, Cinquegrana A, Cavalli R, Vaira M. A Phase I Dose Escalation Study of Oxaliplatin, Cisplatin and Doxorubicin Applied as PIPAC in Patients with Peritoneal Carcinomatosis. Cancers (Basel). 2021 Mar 3;13(5):1060.
Willaert W, Van de Sande L, Van Daele E, Van De Putte D, Van Nieuwenhove Y, Pattyn P, Ceelen W. Safety and preliminary efficacy of electrostatic precipitation during pressurized intraperitoneal aerosol chemotherapy (PIPAC) for unresectable carcinomatosis. Eur J Surg Oncol. 2019 Dec;45(12):2302-2309.”
6) Please discuss the most important research needs which should be addressed in the future in this dynamic field.
We added a paragraph in the “conclusion” part.
“The most important need for research should be focused on two cornerstones. Further in-vitro and in-vivo phase-I trials to evaluate efficacy of other drugs and drug formulations and on randomized clinical phase-III trials comparing PIPAC and standard-of-care systemic chemotherapy in the first-line for peritoneal metastases. These future results, will define the true impact of PIPAC in the context of peritoneal metastases.”

Reviewer 4 Report
This is a descriptive study that includes two parts. In the first part, the authors presented the data of all patients in Germany undergoing PIPAC therapy in 2019. In the 2nd part, the authors showed the data of 44 patients with chemotherapy-refractory peritoneal metastases from gastric cancer undergoing PIPAC-therapy. In the end, the authors concluded that PIPAC is a safe and feasible procedure with low in-hospital morbidity and mortality, and PIPAC in the palliative and the chemorefractory setting is an appealing approach for patient management in the future.
Overall, as several important clinical information was lacking, I do not think the results of this study could convincingly support the conclusion.  At least, the authors have to add a comparison group and examine the morbidity and mortality between groups.
In addition, several points need to be further addressed,
- The abbreviation of PIPAC appeared in the summary and the abstract without explanation, and this will make the abstract difficult to understand.
- In the data of the nationwide German diagnosis-related group (DRG) statistics, more than 80 % of cancer origin was unknown, while this information is key for the analysis. Because different tumor origin is associated with different therapeutic regimens, complications, and outcomes. Unknown cancer origin indicates the data are mixed and could not reach a convincing conclusion.
- Cytoreductive surgery is important, but the study lacks this important information.
- The authors said that PIPAC has low in-hospital morbidity and mortality, but what is the comparison group? without an appropriate comparison group, the authors could not say that the morbidity and mortality are low or high.
Author Response
Reviewer#4:
This is a descriptive study that includes two parts. In the first part, the authors presented the data of all patients in Germany undergoing PIPAC therapy in 2019. In the 2nd part, the authors showed the data of 44 patients with chemotherapy-refractory peritoneal metastases from gastric cancer undergoing PIPAC-therapy. In the end, the authors concluded that PIPAC is a safe and feasible procedure with low in-hospital morbidity and mortality, and PIPAC in the palliative and the chemorefractory setting is an appealing approach for patient management in the future.
Overall, as several important clinical information was lacking, I do not think the results of this study could convincingly support the conclusion.  At least, the authors have to add a comparison group and examine the morbidity and mortality between groups.
In addition, several points need to be further addressed,
- The abbreviation of PIPAC appeared in the summary and the abstract without explanation, and this will make the abstract difficult to understand.
Thank you for that remark. We changed the title and the abbreviation PIPAC is explained.
“Current medical care situation of patients in Germany undergoing Pressurized Intraperitoneal Aerosol Chemotherapy (PIPAC)”
- In the data of the nationwide German diagnosis-related group (DRG) statistics, more than 80 % of cancer origin was unknown, while this information is key for the analysis. Because different tumor origin is associated with different therapeutic regimens, complications, and outcomes. Unknown cancer origin indicates the data are mixed and could not reach a convincing conclusion.
This is indeed a major limitation of this data set. This data set is NOT a prospectively maintained database which includes all the interesting variables we are interested in. The data only contains the most basic information which are necessary for cost refund calculation. The vast majority of hospitals use the ICD-10 code C78.6 (secondary malignancies of the (retro-)peritoneum ) as main diagnosis and not the ICD 10 code which would reflect tumor origin, because reason for admission is not the primary tumor but peritoneal mets. This is the reason why in the vast majority of cases we do not know the primary tumor origin.
The remark of yours, that data are mixed and we cannot reach a convincing conclusion is not completely true because we did not calculate the survival for this cohort. If we had done so your remark would have been absolutely correct.
- Cytoreductive surgery is important, but the study lacks this important information.
Sorry, but I do not get the point of your question. In the context of PIPAC no CRS is performed.
- The authors said that PIPAC has low in-hospital morbidity and mortality, but what is the comparison group? without an appropriate comparison group, the authors could not say that the morbidity and mortality are low or high.
Thank you for your remark. As PIPAC is a unique surgical/chemotherapeutic treatment option for patients with refractory peritoneal metastases, there is no comparison group available. The statement of ours that the complication rate is low has to be interpreted in the context of the M&M profile of other surgical interventions.
